# Polymer nanoparticles pass the plant interface

Sam J. Parkinson[1], Sireethorn Tungsirisurp[2,3], Chitra Joshi[2], Bethany L. Richmond[2], Miriam L. Gifford[2], Amrita Sikder[1], Iseult Lynch[4], Rachel K. O'Reilly[1] ✉ & Richard M. Napier[2] ✉

As agriculture strives to feed an ever-increasing number of people, it must also adapt to increasing exposure to minute plastic particles. To learn about the accumulation of nanoplastics by plants, we prepared well-defined block copolymer nanoparticles by aqueous dispersion polymerisation. A fluorophore was incorporated via hydrazone formation and uptake into roots and protoplasts *of Arabidopsis thaliana* was investigated using confocal microscopy. Here we show that uptake is inversely proportional to nanoparticle size. Positively charged particles accumulate around root surfaces and are not taken up by roots or protoplasts, whereas negatively charged nanoparticles accumulate slowly and become prominent over time in the xylem of intact roots. Neutral nanoparticles penetrate rapidly into intact cells at the surfaces of plant roots and into protoplasts, but xylem loading is lower than for negative nanoparticles. These behaviours differ from those of animal cells and our results show that despite the protection of rigid cell walls, plants are accessible to nanoplastics in soil and water.

One of the most significant scientific advances of the 20th century was the development of synthetic polymeric materials, specifically plastics. Applications using plastics have increased exponentially, such that there is mounting concern over the contamination of marine and terrestrial environments by primary (intentionally added, e.g., as microcapsules for controlled release of agrochemicals) or secondary (breakdown products of larger plastics) micro- and nano-sized plastic particles[1,2], with a Food and Agriculture Organization report emphasising the growing danger of plastics in soils[3]. Studies on the consequences of plant exposure to nanoparticles have focused on hard particles, mostly metal-based nanoparticles plus a few dealing with structured carbon[2,4–8], or chemically crosslinked polymers such as polystyrene and polyethylene beads[9], and so-called superabsorbent polymers utilised for soil remediation[10]. However, little is known about whether or not soft, polymer-based nanoparticles will be taken up by plants, or on their transit in plants if they are accumulated. Given the increasing load on agricultural environments, from the breakdown products of waste plastics and, ironically, the development of soft nano- and microplastics as precision delivery vehicles for agrochemicals[11–15], it is important to evaluate if these might be taken up by plants and to determine the physico-chemical properties that govern or limit uptake of these materials. Given the transition underway currently towards bio-sourced and biodegradable polymers for agricultural mulches and delivery systems, as well as plastic wastes reaching agricultural soils, model systems are needed to track and assess the implications of soft polymer nanoparticles, whose flexile structures and environmental responsiveness may allow them to be taken up and distributed in plants more effectively than their stiffer, more crystalline conterparts[16].

Typically, traditional polymer self-assembly techniques (thin-film rehydration, solvent-switch) have been used to generate polymer nanoparticles to act as agrochemical delivery vectors. These

[1]School of Chemistry, University of Birmingham, Edgbaston, Birmingham B15 2TT, UK. [2]School of Life Sciences, University of Warwick, Coventry CV4 7AL, UK. [3]Department of Analytical, Environmental and Forensic Sciences, Faculty of Life Sciences and Medicine, King's College London, London SE1 9NH, UK. [4]School of Geography, Earth and Environmental Sciences, University of Birmingham, Edgbaston, Birmingham B15 2TT, UK. ✉e-mail: r.oreilly@bham.ac.uk; richard.napier@warwick.ac.uk

techniques have limited scalability and reproducibility and it is not possible to control particle size for similar polymer formulations. To be able to generate model systems for testing soft nanoparticle uptake into plants, a self-assembly technique that allows for controllable nanoparticle sizes to be synthesised is required. Polymerisation-induced self-assembly (PISA) offers controlled soft plastic nanoparticle synthesis using block copolymers[17]. Nanoparticle size, for example, can be managed by tuning the hydrophilic-hydrophobic block ratio. Further, PISA's compatibility with a range of different polymerisation techniques such as reversible addition–fragmentation chain transfer (RAFT)[18,19], atom transfer radical polymerisation[20,21] and ring opening metathesis polymerisation (ROMP)[22,23] allows for a wide range of surface functionalities to be imparted to nanoparticles.

Herein, we evaluate the uptake of a series of soft plastic polymeric nanoparticles into *Arabidopsis* roots. Both intact roots and isolated, cell wall-free root protoplasts have been incubated with nanoparticles of varying small sizes ($D_h = 20{-}100$ nm) and surface functionalities (cationic, neutral, and anionic). The nanoparticles were synthesised using RAFT-PISA and the chemical attachment of a fluorophore allowed for their visualisation within plant tissues and cells using confocal microscopy. We demonstrate the potential of PISA for screening the impacts of systematically varying key physicochemical characteristics (size, surface charge, shape, rigidity, hydrophobicity etc.) of soft polymeric nanoparticles as part of a safe-by-design approach for precision agriculture and greener alternatives to the current "plasticulture".

## Results

### Soft plastic nanoparticle synthesis and characterisation

Initially, four different macromolecular chain transfer agents (mCTAs): poly(dimethyl acrylamide) (PDMAm), poly(acrylic acid) (PAA), poly([2-(methacryloyloxy)ethyl]trimethylammonium chloride) (PQDMAEMA) and poly([2-(methacryloyloxy)ethyl]dimethyl-(3-sulfopropyl)ammonium hydroxide) (PDMAPS) were synthesised via RAFT aqueous polymerisation to allow for different surface charges to be imparted to the self-assembled polymeric nanoparticles (Fig. 1, Supplementary Table 1). Polymeric nanoparticles were then synthesised via RAFT aqueous dispersion polymerisation and each mCTA was chain exten-

ded with diacetone acrylamide (DAAm). For uncharged, neutral nanoparticles PDAAm degree of polymerisations (DP) of 50, 100, and 200 were targeted, the products giving hydrodynamic diameter ($D_h$) values of 23 nm, 37 nm, and 83 nm, respectively, by DLS analysis (Supplementary Fig. 1a) and spherical morphologies were confirmed by TEM (Supplementary Fig. 2).

During the synthesis of negative nanoparticles, it was necessary to screen the negative charges between the PAA chains by introducing PDMAm to the polymerisation reaction of DAAm. The presence of PAA and PDMAm mCTAs at a 1:9 ratio minimised the repulsion of the core acidic groups whilst maintaining an overall negative charge in the surface of these nanoparticles. The neutral PDMAm chains help to screen the negative charges between each PAA chain. With this mCTA mixture, PDAAm DPs of 50, 100, and 200 were again targeted and DLS indicated $D_h$ values of 22 nm, 50 nm, and 98 nm, respectively (Supplementary Fig. 1b).

For positively charged nanoparticles, no nanoparticles were observed for PDAAm DP of 50, and it was assumed that the final polymer was still hydrophilic, and that the DP was not great enough to introduce amphiphilicity to the polymer chain and, hence, induce nanoparticle assembly. For DPs 100 and 200, nanoparticles were observed with $D_h$ values of 28 nm and 40 nm respectively (Supplementary Fig. 1c). Finally, a zwitterionic nanoparticle ($D_h = 30$ nm, Supplementary Fig. 1d) was synthesised for comparison to the neutral PDMAm-based nanoparticles.

To visualise these nanoparticles under confocal microscopy, a fluorophore was incorporated into their core. This fluorophore required an excitation wavelength away from the autofluorescence present in plant cells[24]. For this reason, BODIPY, which also offers a high fluorescence quantum yield, was selected. Attachment was achieved by a hydrazone bond between the BODIPY hydrazide group and the carbonyl groups present in the PDAAm chains (Supplementary Fig. 3). Whilst hydrazone bonds are inherently dynamic in nature, they are stable and previous reports have shown that the use of hydrazone bonds to cross-link PDAAm chains increases particle stability[25].

Attachment of the fluorophore was confirmed for the neutral and negative nanoparticles using size exclusion chromatography by monitoring UV-Vis absorbance (excitation wavelength of BODIPY = 490 nm, Supplementary Fig. 4).

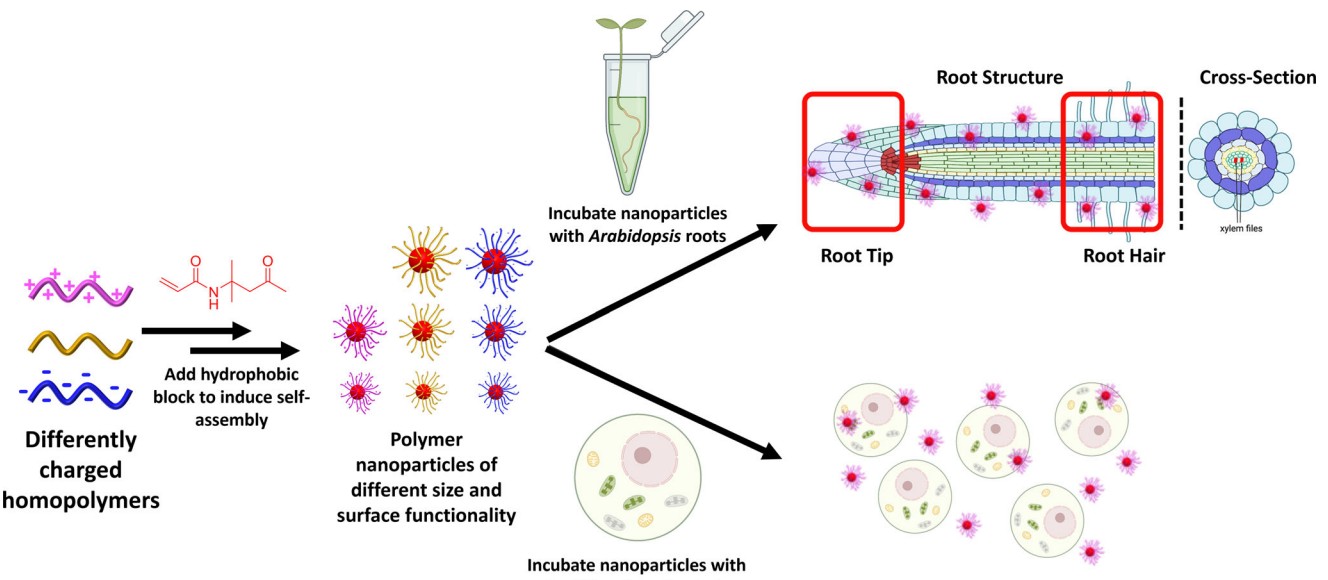

**Fig. 1 | Synthesis of polymeric nanoparticles and subsequent uptake pathways explored.** The charges associated with each class of nanoparticle are colour coded throughout the manuscript: magenta = positively charged; gold = neutral; blue = negatively charged. This figure was created with Biorender.com.

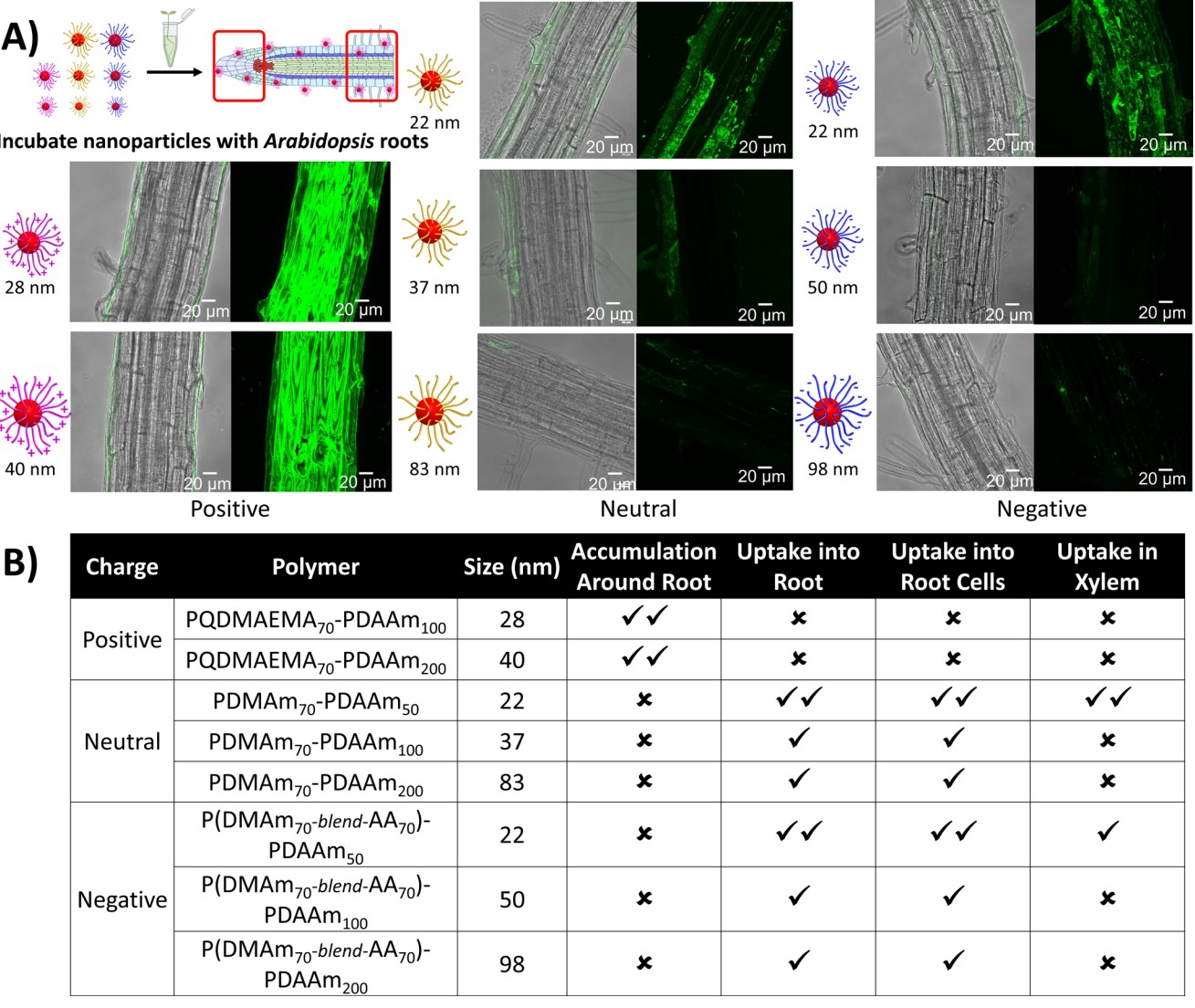

**Fig. 2 | Certain nanoparticles penetrate intact roots. a** Confocal microscopy images for the penetration and distribution of polymeric nanoparticles in *Arabidopsis* root hair zones after one hour of treatment in nanoparticle solution (1 mg/ml) at room temperature. **b** Summary table of the different levels of root penetration observed. Good (✓✓), poor (✓), or none (✗). Penetration and accumulation were evaluated under a ZEISS 880 LSM. Maximum Z projections in 488 nm laser channel were analysed alongside the Z-slices and merged with brightfield images using ImageJ software. Scale bar = 20 µm. The images are representatives of experimental replicates (*n* = 3). Part of this figure was created with Biorender.com.

## Visualising uptake and accumulation in intact roots

To investigate uptake of the differently sized and charged soft nanoparticles into plants, 5-day-old *Arabidopsis thaliana* (Col-0) seedlings were incubated with each nanoparticle preparation for an hour. The accumulation (uptake and local distribution into cells) of polymer nanoparticles into the roots was monitored using confocal microscopy at both the tip, including the apex and meristematic regions (Supplementary Fig. 5), and further from the apex in the more mature, root hair zone (Fig. 2). Plant cell walls might be considered effective physical barriers to ingress, but accumulation was seen, and penetration was found to be inversely proportional to particle size (Fig. 2). Small, uncharged nanoparticles (~20 nm) could cross the cell wall barrier and were readily taken up by the roots. A significant accumulation of neutral nanoparticles was seen inside root hairs and epidermal cells as well as at the lateral root cap and columella (Supplementary Fig. 5), indicating successful penetration through primary plant cell walls. These neutral nanoparticles are likely to have no electrostatic interactions with cell walls allowing them to diffuse into cells to the extent that some fluorescence was also seen in the xylem files of the vascular system (Fig. 2).

There was a marked reduction in penetration and accumulation as the neutral nanoparticle size increased, with no fluorescence found in the xylem files for 40 nm particles or above. A moderate accumulation of fluorescence for these larger particles was detected only in the shedding lateral root cap cells and in occasional epidermal cells in the root hair zone (Fig. 2). These observations were extended with our zwitterionic nanoparticles which showed similar uptake and accumulation profiles to neutral particles (Supplementary Fig. 6).

The smallest negatively charged nanoparticles also accumulated into epidermal cells and root hairs (Fig. 2) and accumulation was observed in shedding lateral root cap cells at the root tip (Supplementary Fig. 5). However, no fluorescence was detected along the xylem files in the vascular system indicating that these particles are far less mobile across and between cell layers than the neutral particles. Negatively charged nanoparticles are likely to exhibit electrostatic interactions with acidic polysaccharides and associated ions in the cell

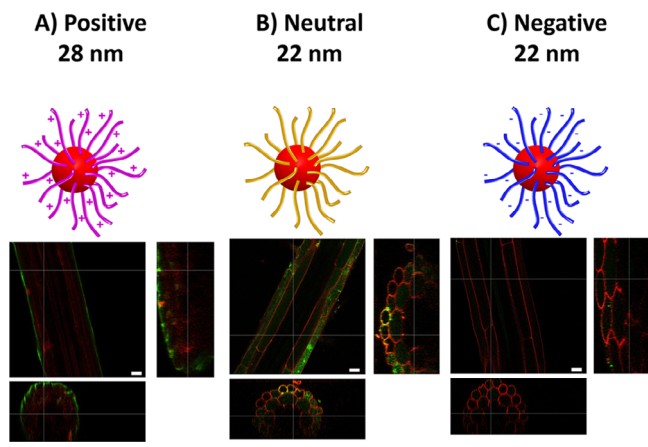

**A) Positive**
**28 nm**

**B) Neutral**
**22 nm**

**C) Negative**
**22 nm**

**Fig. 3 | Root cross-sections illustrate nanoparticle accumulations.** Cross-sections of root hair zones after uptake of small nanoparticles with (**a**) positive, (**b**) neutral, and (**c**) negative surface charges, respectively. Propidium iodide staining was used to outline the cellular structure (red) while the nanoparticles are in green, and areas of co-localisation appear yellow. ImageJ software was used to generate the orthogonal views from the Z-stack images. Scale bar = 20 μm. The images are representatives of experimental replicates (*n* = 3).

wall, affecting their accumulation and uptake[7,26]. As with our neutral particles, increasing particle size inhibited uptake.

Positively charged nanoparticles were observed all around the roots where they appeared to sheathe the surface. This may be due to attractive electrostatic interactions with the plant cell wall, and it appears to prohibit further passage of particles into or past the adjacent cells. This surface coating of roots has also been reported for hard nanoparticles as a function of their surface charge[7,26–29]. Our results are summarised in Fig. 2b.

To help assess the degree of penetration of the smallest nanoparticles, orthogonal views of the confocal images were constructed. Propidium iodide (PI) staining was used to reveal the cell walls and resolve the tissue structures[30]. These orthogonal views confirmed a sheet-like layer of positive nanoparticles (Fig. 3a) coating the surface of the root, uptake of small neutral nanoparticles (Fig. 3b) into both epidermal and cortical cells, and only a little accumulation of negative nanoparticles (Fig. 3c).

### Uptake into protoplasts
Having established that some polymer nanoparticles could penetrate into plant organs and, hence, past primary cell walls, we investigated their uptake into protoplasts (Fig. 4). Protoplasts are plant cells that have had their cell walls removed enzymatically. BODIPY fluorescence emission was observed on damaged cells and cell debris regardless of the nanoparticle surface chemistry, although negative nanoparticles did show a reduced interaction with cell debris. The uptake or adsorption onto cell debris declined with increasing particle size. Interestingly, the same pattern of accumulation into the healthy (round and spherical) protoplasts was seen as in intact roots, with small neutral (Fig. 4A middle) and small zwitterionic (Supplementary Fig. 7) nanoparticles penetrating readily. The smallest negatively charged nanoparticles accumulated in healthy protoplasts, but larger ones did not. Positively charged particles were not observed in healthy cells, but as in intact tissues they did collect around protoplasts as well as on debris. The data suggest that positive surface charge inhibits penetration through intact plant plasma membranes as well as through the cell wall.

We may compare our data from plant exposure to soft nanoparticles to that for hard nanoparticles and to data for animal cells. In plants, gold nanoparticles, regardless of charge, were found not to pass the cell wall in *Arabidopsis*[7], but that both positive and negatively

charged particles did enter protoplasts. Positively charged gold nanoparticles bound rapidly to the plasma membrane of tobacco protoplasts and were internalised efficiently by clathrin-mediated endocytosis, whereas negatively charged particles bound to fewer areas of membrane and were internalised by other mechanisms[31]. Further, metal oxide nanoparticles have been found to induce changes in gene expression in *Arabidopsis*[32]. Clearly, plant cells can internalise nanoparticles if they reach the plasma membrane, and our data (Fig. 4b) show that neutral particles will be internalised and accumulate rapidly.

Ready access to the plasma membrane of animal cells is similar to the case for protoplasts. As with plants, most studies have concentrated on hard particles of gold, silica, quantum dots, and carbon nanotubes. Most nanoparticles will be taken up by animal cells, with the most effective sizes being between 40 nm, and 70 nm[33]. Uptake in this size range is dominated by clathrin-mediated endocytosis and generally positively charged particles are internalised fastest and most effectively. Although some cell types preferentially internalise anionic particles[34], protein-corona binding typically results in a negative surface charge[35]. There is some evidence that soft, small nanoparticles are internalised into cancer cells more effectively than hard particles of a similar size[36]. Our data show clearly that there are fundamental differences between soft nanoparticle interactions with plant and animal cell systems even once the protective cell wall is breached.

### Time dependence of nanoparticle uptake
In all uptake experiments performed so far roots were incubated with nanoparticles for 1 h. In the interest of examining how uptake might change over time, time series experiments were performed using the smallest nanoparticle of each functionality with incubation times ranging from 1 h to overnight (Fig. 5).

As previously noted, positive nanoparticles coated the roots and this persisted through all time points (Fig. 5). The accumulation of negative nanoparticles, which was low initially, was observed to increase with time. In particular, fluorescence was detected in the xylem files after 2.5 h, becoming prominent after 5 h, and continued to increase with the overnight incubation time. These nanoparticles will thus be delivered throughout the plant in the transpiration stream. No signs of separation of the covalently-coupled BODIPY fluorophore from nanoparticles were detected when incubated in apoplastic fluid (Supplementary Fig. 8).

The accumulation of neutrally charged nanoparticles into cells on or near the root surface peaked within an hour, after which uptake appeared to decline. After 24 h there was fluorescence in the paired xylem vessels, although the signal was weak (Fig. 5). Similar observations were made for our zwitterionic nanoparticles without the acute early acute accumulation phase (Supplementary Fig. 9). This early burst of accumulation of neutral nanoparticles might indicate that the plants are responding to the influx and reacting to eject or block entry as with the immune response to restrict viral distribution in plants[32,37]. We did test for transcriptome responses by monitoring the expression of a small set of abiotic and biotic stress genes (Supplementary Table 3; Supplementary Fig. 10)[32]. Results indicated a general reaction to all nanoparticle types but no immediate early response specifically to neutral particles was found within the panel of genes tested. Within the period of investigation, no damage to plants was observed under any of the treatments.

### Discussion
In summary, we have reported the uptake of polymeric nanoparticles with different sizes and surface functionalities into *Arabidopsis* root cells. By utilising RAFT-PISA, we were able to easily alter the size and functionality of these nanoparticles simply by increasing the core block DP or using a different hydrophilic macro-CTA. Attachment of a BODIPY-based fluorophore, via hydrazone bond formation with the

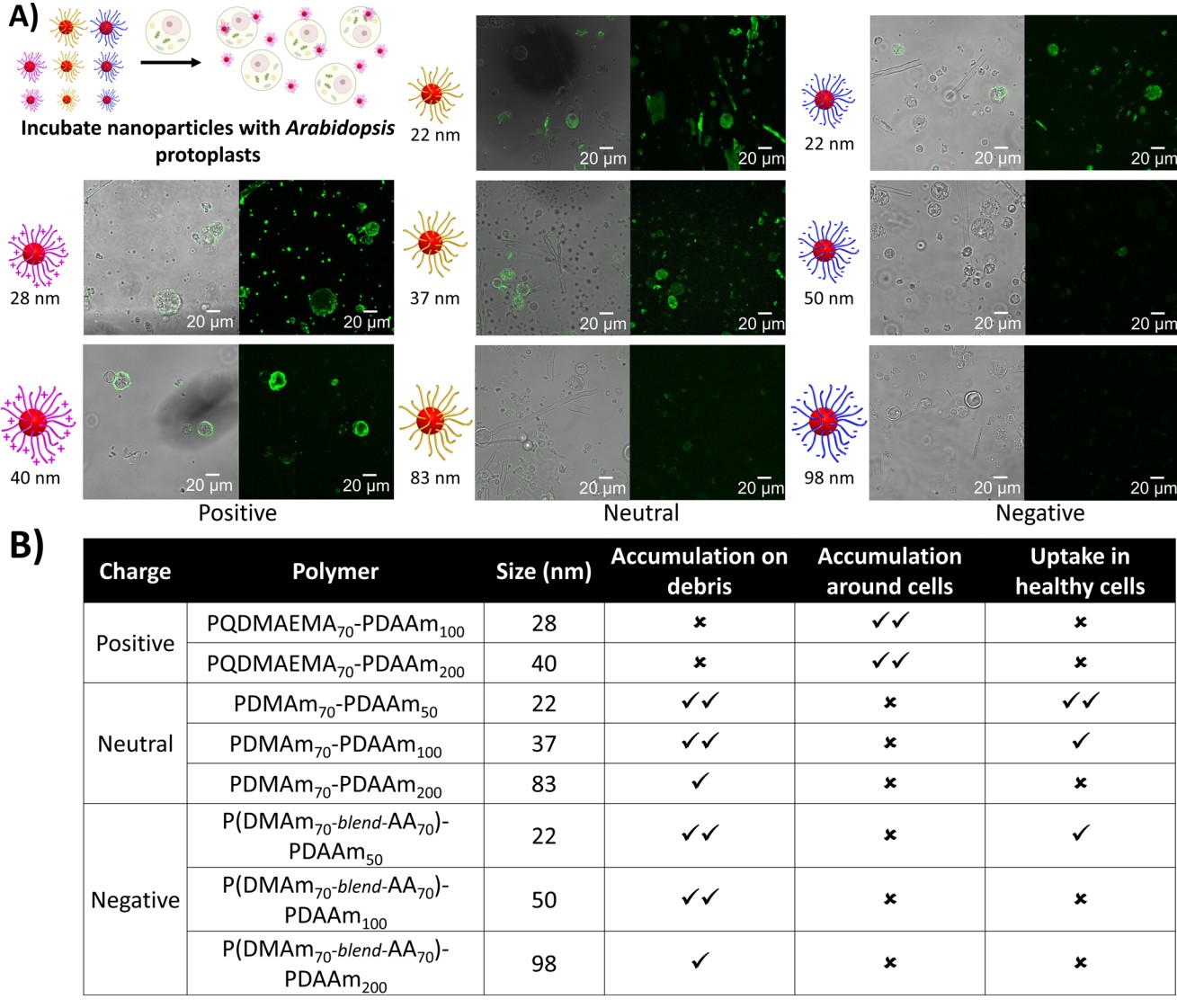

**Fig. 4 | Certain nanoparticles penetrate into root cell protoplasts. a** Confocal images for the penetration and distribution of polymeric nanoparticles with *Arabidopsis* protoplasts. **b** Summary table of the different levels of penetration observed. Good (✓✓), poor (✓), or none (✗). Penetration and accumulation were evaluated using a ZEISS 880 LSM. Maximum Z projections in the 488 nm laser channel were analysed alongside the Z-slices and merged with brightfield images using ImageJ software. Scale bar = 20 µm. The images are representatives of experimental replicates ($n = 3$). It was noted that the smallest positively charged nanoparticles appeared to have a cytotoxic effect with an increase in visible cell debris and rough, asymmetric cells compared to other nanoparticle systems. Part of this figure was created with Biorender.com.

poly(diacetone acrylamide) cores, allowed visualisation of these particles within plant tissues. Uptake into *Arabidopsis* roots and into root protoplasts was found to be inversely proportional to nanoparticle size. This differs from animal cells for which there is an optimum size range ~50 nm which equates to the mid-sized particles used here.

Given the characteristics of the cellulose matrix in plant primary cell walls the size exclusion limit reflects, in part, natural pore size limitations. However, it is notable that others have found that even very small hard nanoparticles are excluded[7,33], suggesting that we need to evaluate soft polymer nanoparticles separately and with care. This seems particularly to be the case for soft particles with neutral or a negative charge. Uncharged soft plastic nanoparticles penetrated into superficial intact cells readily, and soft particles with a negative charge penetrated more slowly, but this low uptake rate persisted leading to substantial loading of the transpiration stream over time (Fig. 5). It is somewhat surprising, perhaps, that once past the cell walls neutral and negatively charged nanoparticles penetrate through the plasma membrane and accumulate in the cytoplasm given the considerable turgor pressure exerted by plant cells, but our observations on protoplasts are consistent with data for gold, other metal and other hard nanoparticles and the principal mechanism appears to be clathrin-mediated endocytosis, as in animal cells[7,31,38,39].

The treatments given in this work were short, but penetration of charged polystyrene nanoparticles into plants through their roots has been reported for plants grown for a week in the treatments[40]. By using fluorescently labelled nanoparticles ($D_h = 200$ nm), amine-coated (positive) nanoparticles were found to accumulate primarily in the root epidermis of *Arabidopsis* plants, whilst sulphonate (negative) coated particles could be observed in deeper root tissues. Thus, despite their comparatively large sizes, these particles did gain access and accumulate over long periods. Worryingly, similar but smaller ($D_h = 55–71$ nm) particles impaired plant development[40]. Foliar loading of small, charged, gadolinium-loaded star polymer nanoparticles has also been studied three days after treatment[37]. Most treatments included a surfactant wetting agent to combat the hydrophobicity of the plant cuticle and this is likely to have affected uptake and

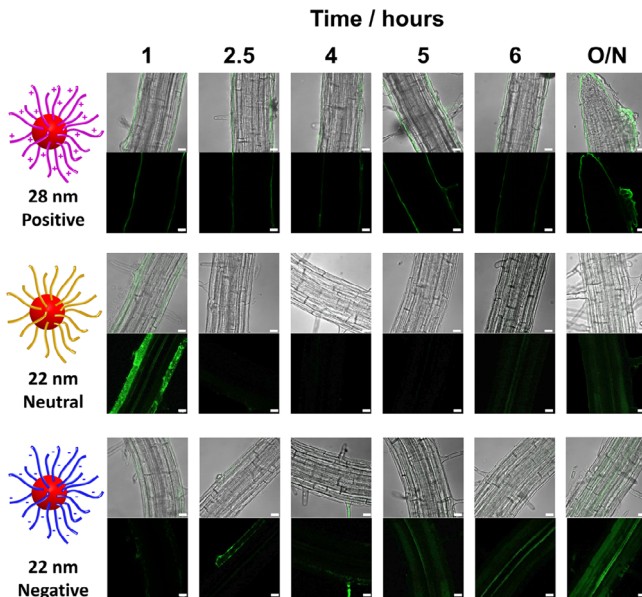

**Fig. 5 | Accumulation of nanoparticles over time.** Confocal images for the penetration and accumulation of the smallest polymeric nanoparticles in *Arabidopsis* root hair zones over time. Penetration and accumulation were evaluated using a ZEISS 880 LSM. The images are representatives of experimental replicates (*n* = 3). Maximum Z projections in the 488 nm laser channel were analysed alongside the Z-slices and merged with brightfield images using ImageJ software. Scale bar = 20 μm.

penetration. Once inside the tissues, both symplastic (smaller particles) and apoplastic (larger particles) transport was reported for negatively charged polymers. The results reported here of immediate early ingress of soft plastic nanoparticles and of reactions to this challenge are consistent with the few reports of long-term treatments, which have been generally of many days[32,40,41]. This work illustrates that naïve plant tissues are not protected from soft plastic nanoparticles by their cell walls and that for the smallest particles accumulation is rapid and persistent.

Sustainable arable agriculture must provide sufficient nutrition for the growing global population. To date, it has generally managed to increase productivity in line with growing need. As with other parts of the biosphere, the quality of available land is decreasing due to climate change, pollution, and over-fertilisation leading to low nutrient use efficiency, and so we need to ensure that we understand, as far as possible, new and increasing threats to crop production. Nanoplastics might be a threat previously overlooked, and this work provides a foundation for continued careful evaluation of these materials. At the same time, we should recognise that nanopolymers might also provide new opportunities for reducing agrochemical inputs through precision application technologies. To this end, we must ensure that the materials developed are not damaging and that they reach their intended destinations with greatest efficiency. If nanoparticles are to be used as vectors for crop protection, for example, the findings reported here of preferred transit to the xylem of small (<40 nm), neutral and negatively charged nanoparticles is instructive. However, the same properties potentially make these kinds of nanoparticles most likely to pass into the food system. Armed with deeper understanding it becomes possible to tailor the application rates to optimise delivery and reduce residues, for example by further designing the soft nanoparticles to be biodegradable or hydrolysable after they have performed their agricultural function.

Overall, we have highlighted some of the different factors that can be used to modulate nanoparticle uptake and demonstrated RAFT-

PISA as a highly versatile platform for systematic exploration of the influence of polymeric particle properties on their accumulation and localisation in plants. Subsequent work will consider other parameters such as particle stiffness (e.g., by varying the cross-link density) and impact of hydrolysable groups on retention of the nanoparticles. We believe this work begins to provide a basis for understanding how different polymeric nanoparticles will interact with plants in real-world settings.

## Methods
### Materials
Dimethyl acrylamide (DMAm, 99 %; Sigma Aldrich 274135), Acrylic Acid (AA, 99 %; Sigma Aldrich 147230), (Methacryloyloxy)ethyl] trimethylammonium chloride (QDMAEMA, 75 % in $H_2O$; Sigma Aldrich 408107), (Methacryloyloxy)ethyl]dimethyl-(3-sulfopropyl)ammonium hydroxide (DMAPS, 95 %; Sigma Aldrich 537284), 4,4′-azobis(4-cyanovaleric acid) (ACVA, 99%; Sigma Aldrich 11590), 1-ethyl-3-(3-dimethylaminopropyl)carbodiimide (EDC, 98%; Sigma Aldrich 341006), deuterated methanol ($CD_3OD$, 99.8%; Sigma Aldrich 151947-10G-GL) and deuterium oxide ($D_2O$, 99.9%; Sigma Aldrich 151882) were purchased from Sigma Aldrich (UK). DAAm (99%) was purchased from Alfa Aesar (UK; A15940.30). 4,4-Difluoro-5,7-dimethyl-4-bora-3a,4a-diaza-s-indacene-3-propionic acyl hydrazide (BODIPY FL, 99 %) was purchased from ThermoFisher Scientific (UK; D2371). 4-((((2-Carboxyethyl)thio)carbonothioyl)thio)−4-cyanopentanoic acid (BM1433, 95%) was purchased from Boron Molecular (USA; BM1433).

For protoplast preparation, cellulase "Onozuka RS" was purchased from Melford biolaboratories (Duchefa; C8003.0005) and pectolyase from Sigma Aldrich (P3026).

### $^1H$ NMR spectroscopy
$^1H$ NMR spectra were recorded at 300 MHz on a Bruker DPX-400 spectrometer in either $D_2O$ for macro-CTA synthesis or $CD_3OD$ for all nanoparticle syntheses.

### Size exclusion chromatography
SEC measurements were performed on a Varian 390-LC-Multi detector suite system fitted with Refractive Index (RI) and ultraviolet (UV) detectors ($\lambda$ = 309, 490 nm) equipped with a PLGel 3 μm (50 × 7.5 mm) guard column and two PLGel 5 μm (300 × 7.5 mm) mixed-D columns using DMF with 5 mM $NH_4BF_4$ at 50 °C as the eluent at a flow rate of 1.0 mL min⁻¹. SEC data were calibrated against polystyrene standards and analysed using Cirrus v3.3 software.

### Transmission electron microscopy (TEM)
TEM was performed using a JEOL 2000FX or JEOL 2100FX at 200 kV. TEM solution was typically made up at 0.1 mg mL⁻¹ in water. Then, 10 μL of sample solution was dropped onto a carbon/formvar-coated copper grid placed on filter paper. After removing excess liquid, 10 μL of a 1% uranyl acetate solution was dropped onto the grid and left to dry.

### Synthesis of hydrophilic macro-chain transfer agents (macro-CTAs)
A typical synthesis of a PDMAm₇₀ macro-CTA was as follows: dimethyl acrylamide (10 g, 100 mmol, 70 eq.), BM1433 (0.44 g, 1.4 mmol 1 eq.), ACVA (0.04 g, 140 μmol 0.1 eq.) were added to a round bottom flask and dissolved in water (24 mL) to give a 30% w/w reaction solution. A stirrer bar was added and then the flask was sealed and sparged with nitrogen for 20 min. The sealed flask was then immersed in an oil bath at 70 °C and left for 120 min after which it was removed from the oil bath and quenched by exposure to oxygen. Samples were then taken for $^1H$ NMR and SEC analysis followed by purification by dialysis and then lyophilisation to yield a yellow powder. The same procedure was followed for all other macro-CTAs.

## Synthesis of diblock copolymer nanoparticles

A typical synthesis of a PDMAm$_{70}$-PDAAm$_{50}$ was as follows: dia-cetone acrylamide (1 g, 6 mmol, 50 eq.), PDMAm$_{70}$ mCTA (0.85 g, 120 µmol 1 eq.), ACVA (3 mg, 12 µmol 0.1 eq.) were added to a round bottom flask and dissolved in water (7.4 mL) to give a 20% w/w reaction solution. A stirrer bar was added and then the flask was sealed and sparged with nitrogen for 20 min. The sealed flask was then immersed in an oil bath at 70 °C and left for 120 min after which it was removed from the oil bath and quenched by exposure to oxygen. Samples were then taken for $^1$H NMR and SEC analysis. No further purification was performed for further experiments. The same procedure was followed for all other diblock copolymer nanoparticles (see Supplementary Table 1).

## BODIPY fluorophore attachment to polymer nanoparticles

A typical attachment of BODIPY FL to PDMAm$_{70}$-PDAAm$_{50}$ nano-particles was as follows: EDC (240 µg, 0.6 µmol, 1 eqv) was added to PDMAm$_{70}$-PDAAm$_{50}$ (100 mg, 0.6 µmol, 1 eq.) in water (10 mL) and stirred for 5 min. BODIPY FL (19 µg, 0.06 µmol, 0.1 eqv) dissolved in DMSO (15 µL) was then added and the solution was left to stir over-night. The solution was then purified by spin centrifugation against a 3k MWCO membrane. A sample was taken for SEC to confirm attach-ment of BODIPY to the polymer chains. The same procedure was fol-lowed for all other nanoparticles.

## Arabidopsis thaliana root preparation

*A. thaliana* ecotype Columbia-0 was used for all nanoparticle uptake experiments. The seeds were surface-sterilised using successive washes in 10% Bleach or 70% ethanol prior to sowing onto sterile ½ strength Murashige and Skoog (MS) medium (2.2 g/L MS supple-mented with B5 vitamins, 1% sucrose, 1% agar, pH 5.8). Seeds were stratified in the dark for two days at 4 °C prior to germination for 5 days at 22 °C with 12 h daylength.

## Protoplast isolation

Protoplast solution (600 mM Mannitol, 2 mM MgCl$_2$, 2 mM CaCl$_2$, 10 mM KCl, 2 mM MES, 0.1% w/v bovine serum albumin) was prepared and adjusted to pH 5.5 with Tris-HCl, 0.2 µm filtered, and stored at −20 °C until use. To isolate protoplasts, 5-day-old *Arabidopsis thaliana* roots were transferred into enzyme solution (1.5% w/v Cellulase RS, 0.1% w/v Pectolyase in protoplast solution) and chopped finely using a sterile blade. The solution with chopped roots was transferred to a 35 mm round Petri dish and incubated in the dark at 25 °C with con-stant agitation for at least 2 hr. The resulting cell suspension was fil-tered sequentially through 70 µm and 40 µm meshes pre-soaked with enzyme solution. The protoplasts in solution were carefully trans-ferred to polystyrene culture tubes and an equal volume of fresh protoplast solution added, before centrifugation at 300×*g* for 5 min at 4 °C. The supernatant was discarded, the protoplasts resuspended in the same volume of protoplast solution and centrifugation repeated. The protoplasts were resuspended in 500 µL protoplast solution and used directly.

## Confocal microscopy

Intact *Arabidopsis* roots were incubated with nanoparticle samples for an hour before visualisation under fluorescence confocal microscopy. Seedling roots were dipped into 100 microlitres of the nanoparticle solution (all 1 mg/ml in water) or water in a microfuge tube. For con-focal microscopy, root or protoplast samples were mounted on glass slides (1.0–1.2 mm thick) with a long cover glass (22 × 50 mm, 0.16–0.19 mm thick) before visualisation using a Zeiss LSM 880 instrument under ×40 objectives lens with excitation laser at 488 nm and collecting emission at 496−577 nm. The images were stacked and reconstructed by ZEN software and analysed using ImageJ software. After incubation with nanoparticles, roots were also stained with PI for 10 min and propidium fluorescence was visualised with excitation at 561 nm and emission collected from 580−718 nm.

## Quantitative PCR

A set of genes known to respond to abiotic and disease stresses[32] (Supplementary Fig. 10) were selected for reverse transcription quantitative PCR (RT-qPCR). Primers were designed (Supplementary Table 2) and reaction conditions optimised for all steps in the protocol.

*Arabidopsis thaliana* Col-1 seeds were grown on agar plates (half-strength MS salts plus 0.5% sucrose). At seven days old, seedlings were treated with nanoparticles for 90 min, then flash-frozen in liquid nitrogen and stored at −80 °C. There were three experimental treat-ments−positively-, negatively- and neutrally charged nanoparticles− and a control group (water only). Three biological replicates were performed.

20 mg of tissue per sample was homogenised using the Tissue-Lyser II Sample Disruptor (Qiagen). Three sterile metal beads were added to each sample and disrupted for 2 min at 25 Hz frequency, followed by incubation on dry ice for 5 mins, then another disruption at the same settings.

Total RNA was isolated using the Monarch Total RNA Miniprep Kit (New England BioLabs; #T2010) with the manufacturer-recommended on-column DNase treatment. Elution was done in 100 µl nuclease-free water. Nucleic acid concentrations were quantified, and contamination assessed using the NanoDrop ND-1000 Uv-Vis Spectrophotometer (ThermoFisher Scientific). All samples had a 260/230 ratio ≥2 and a 260/280 ratio ≥2, and were stored at −80 °C. RNA integrity was assessed using the 2100 Bioanalyzer Instrument (Agilent); all samples had a RIN number >5. RNA was stored at −80 °C.

1 µg total RNA per sample was reverse transcribed using the Pro-toScript II First Strand cDNA Synthesis Kit (New England BioLabs; E6560L) following manufacturer's instructions in a 20 µL reaction volume with the included Random Primer Mix (60 µM). The resultant cDNA was diluted 10-fold with nuclease-free water and stored at −20 °C.

qPCR reactions were performed in 96-well plates (ThermoFisher Scientific) with a Stratagene Mx3005P system (Agilent Technologies) using SYBR Green JumpStart Taq ReadyMix (Sigma Aldrich; S4438). Reactions were in 10 µl volumes containing 500 nM of each primer, 0.5× SYBR Green mix, and 1 µl of cDNA; each qPCR reaction was run in triplicate. Reactions were performed as follows: 94 °C initial dena-turation for 2 min, preceded by 40 cycles of 94 °C for 15 s and 51 °C or 53 °C for 1 min, denaturation at 95 °C for 1 min, 55 °C for 30 s and a final denaturation step at 95 °C for 30 s.

The constitutively expressed genes *UBIQUITIN10 (UBQ10*, AT4G05320) or *TAP42 INTERACTING PROTEIN OF 41 KDA (TIP41*, AT4G34270) were used as endogenous controls for normalisation. Experimental genes tested assessed were: *FLG22-INDUCED RECEPTOR-LIKE KINASE 1 (FRK1*, AT2G19190), *PHOSPHITE-INSENSITIVE 1 (PHI1*, AT2G21870), *NDR1/HIN1-LIKE (NHL10*, AT2G35980) and *WRKY1* (AT2G04880). All primer specificities were assessed with an in silico specificity screen using Primer-Blast (NCBI). Amplicon sizes are between ~70 to 280 bp. Primers are in gene exons, except the UBQ10 primers which fall in the 3'UTR. All primer pairs had PCR efficiencies of between 90 and 110%.

Data was outputted into MxPro qPCR Software (Agilent). The Cq threshold was set to 1000, and outliers omitted from analysis if deviation from mean Cq was >0.5. The Pfaffl Method of normalisation was used and statistical analysis performed in R Studio.

## Data availability

The authors declare that the data supporting the findings of this study are available within the article and its supplementary information files. Original image files and the quantitative qRT-PCR data have been deposited in the Dryad database https://doi.org/10.5061/dryad.

cc2fqz69d. Any additional details may be requested from the corresponding author (RN).

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

## Acknowledgements

R.K.O., S.J.P., S.T., and R.M.N. acknowledge financial support from the Leverhulme Trust for funding (Grant number RPG-2016-452). A.S. acknowledges funding from the European Union's Horizon 2020 research and innovation programme under the Marie Skłodowska-Curie grant agreement no. 897666. B.L.R. thanks the Biotechnology and Biological Sciences Research Council for a MIBTP studentship and M.L.G. for award BB/P002145/1.

## Author contributions

R.K.O. and R.M.N. designed the research and M.L.G. designed the RT-qPCR work; S.J.P., S.T., C.J., B.L.R., and A.S. performed the research; S.J.P., I.L., R.K.O., and R.M.N. wrote the paper.

## Competing interests

The authors declare no competing interests.
