## [Peer Review File · Nature Communications]

Polymer nanoparticles pass the plant interfaceREVIEWER COMMENTS

Reviewer #1 (Remarks to the Author):

In this study, the authors prepared well-defined block copolymer nanoparticles with a range of different sizes (Dh = 20 - 100nm) and surface chemistries by aqueous dispersion polymerization, and investigated their speed of uptake and distribution within Arabidopsis thaliana by fluorescent labeling. In general, this design is interesting and well organized. However, the uptake and conduction of polymer nanoparticles in plants have been investigated by many scholars and the novelty is not good enough for publication in Nature Communications.

1) The authors only explored the effects of polymer nanoparticles of three different charges on plant uptake; If particles of charges of same charge with different numbers of charges also have different effects on plant uptake?

2) The authors only mention three particles with different charges, and the number of charges of the particles is not tested. We recommend testing a zeta potential.

3) The author mentions that 5-day-old Arabidopsis thaliana (Col-0) seedlings were incubated with each nanoparticle preparation for an hour. It is not clearly expressed how to incubate it.

4) Some references are inappropriate and need to be updated appropriately.

Reviewer #2 (Remarks to the Author):

It is a very well-written and nicely organized manuscript, which has been a joy to read. The study by Parkinson and co-authors sheds light on how plants respond to different charged soft plastic nanoparticles and addresses the hazardous impact of plants' vulnerability to nanoplastic particles in the water or soil. The present research provides important novel pieces of information about the observed higher accumulation of small neutrally charged nanoparticles in root hairs and epidermal cells as well as at the lateral root cap, columella, xylem, and protoplast within an hour, after which uptake appeared to decline due to possible plant immune responses. The small negatively charged nanoparticles were also accumulated into epidermal cells, root hairs, and protoplast but not in the xylem due to the possible electrostatic interactions with acidic polysaccharides in the cell wall, affecting their accumulation and uptake in comparison with neutral ones. Interestingly, the accumulation of negative nanoparticles has increased with time. On the contrary, positive nanoparticles could not penetrate the roots and protoplasts through all time points. This work provides implications for the future strategies for the delivery of nanoscale active ingredients that would be interesting.

Comments:

In all figures, the authors need to provide the used nanoparticle size on the confocal images to help the readers follow up instead of going back and forth between the text and figures.

The authors concluded that small neutrally charged nanoparticles uptake declined due to possible plant immune responses, but the accumulation of negative nanoparticles has increased with time. Why did the plant immune response not respond to negative nanoparticles' accumulation although plants had more time to fight back?

It was worthy to check the effect on plant development and to check the expression of plant defensive genes in response to those nanoparticles. The authors could have used molecular and biochemical tools to prove their hypothesis.

Please find below our commentary on how we have dealt with each of the points made by the two previous referees. I list both the original comments from each referee and our reply.

Richard Napier

REVIEWER COMMENTS

Reviewer #1 (Remarks to the Author):

In this study, the authors prepared well-defined block copolymer nanoparticles with a range of different sizes ($D_h = 20 - 100\text{nm}$) and surface chemistries by aqueous dispersion polymerization, and investigated their speed of uptake and distribution within *Arabidopsis thaliana* by fluorescent labelling. In general, this design is interesting and well organized. However, the uptake and conduction of polymer nanoparticles in plants have been investigated by many scholars and the novelty is not good enough for publication in Nature Communications.

Reply: We thank the referee for recognising strengths in the manuscript. In reply to the point on novelty, we respectfully ask that the following points are taken into consideration: Our nanoparticles are well-defined block copolymer nanoparticles and we refer to these as “soft” nanoparticles because they are distinctly different from for example, heavy metal or crystalline particles – for which we accept and recognise (refs 3 – 10 in the manuscript) prior knowledge; There is a little other work using carbon-based polymers (e.g. Sun et al., 2020 and Zhang et al., 2021), but in these cases plants were grown or treated over many days in the particles, whereas our study looks at naive exposure of whole plants over short time periods (one hour). In one experiment we extended uptake to overnight) to allow an appreciation of the dynamics of accumulation during exposure, another aspect novel to our study. Furthermore, our work aims to highlight the usefulness of polymerisation-induced self-assembly (PISA) as a platform for the development of polymer nanoparticles which can easily control both particle size and surface chemistry. To further highlight this, we have added the following to the manuscript introduction “Typically, traditional polymer self-assembly techniques (thin-film rehydration, solvent-switch) have been used to generate polymer nanoparticles to act as agrochemical delivery vectors. These techniques have limited scalability and reproducibility as well as not being able to control particle size for similar polymer formulations. To be able to generate model systems for testing soft nanoparticle uptake into plants, a self-assembly technique that allows for controllable nanoparticle sizes to be synthesised is required.”

Nevertheless, to respect the point made, we have removed the following from the Abstract “It is not yet known whether plants will accumulate such micro- and nanoplastic materials ...”; and included a fuller discussion on two relevant recent publications in the Discussion.

1) The authors only explored the effects of polymer nanoparticles of three different charges on plant uptake; If particles of charges of same charge with different numbers of charges also have different effects on plant uptake?

Reply: Thank you for this idea. We undertook to test the uptake of the nanoparticles with respect to the surface chemistry of the particles and believe it is fair to report on the work completed. The zeta potentials of our particle set is now shown in Table S1 and the different sizes within charge groups display similar potentials. In response to the referee’s

comment, we synthesised nanoparticles with different charge densities and found that their zeta potential's closely matched those of our original nanoparticle set and we have not assessed uptake for these additional particles. We accept that there are always more parameters to explore, we do not feel that further work on this parameter adds significant additional value to the story presented which relates to surface charge.

2) The authors only mention three particles with different charges, and the number of charges of the particles is not tested. We recommend testing a zeta potential.

Reply: As mentioned in the response to the previous comment, the zeta potentials of all nanoparticles have now been measured and added to table S1 in the supporting information.

3) The author mentions that 5-day-old Arabidopsis thaliana (Col-0) seedlings were incubated with each nanoparticle preparation for an hour. It is not clearly expressed how to incubate it.

Reply: Thank you, these details have now been added to the relevant methods section. The confocal microscopy method section was also moved after the section describing Arabidopsis growth.

4) Some references are inappropriate and need to be updated appropriately.

Reply: We thank the reviewers for pointing this error out and the references have been appropriately updated.

Reviewer #2 (Remarks to the Author):

It is a very well-written and nicely organized manuscript, which has been a joy to read. The study by Parkinson and co-authors sheds light on how plants respond to different charged soft plastic nanoparticles and addresses the hazardous impact of plants' vulnerability to nanoplastic particles in the water or soil. The present research provides important novel pieces of information about the observed higher accumulation of small neutrally charged nanoparticles in root hairs and epidermal cells as well as at the lateral root cap, columella, xylem, and protoplast within an hour, after which uptake appeared to decline due to possible plant immune responses. The small negatively charged nanoparticles were also accumulated into epidermal cells, root hairs, and protoplast but not in the xylem due to the possible electrostatic interactions with acidic polysaccharides in the cell wall, affecting their accumulation and uptake in comparison with neutral ones. Interestingly, the accumulation of negative nanoparticles has increased with time. On the contrary, positive nanoparticles could not penetrate the roots and protoplasts through all time points. This work provides implications for the future strategies for the delivery of nanoscale active ingredients that would be interesting.

Reply: We thank the reviewer for this encouraging review.

Comments:

In all figures, the authors need to provide the used nanoparticle size on the confocal images to help the readers follow up instead of going back and forth between the text and figures.

Reply: We thank the reviewer for highlighting this and have added the nanoparticle size used onto each confocal image for clarity.

The authors concluded that small neutrally charged nanoparticles uptake declined due to possible plant immune responses, but the accumulation of negative nanoparticles has increased with time. Why did the plant immune response not respond to negative nanoparticles' accumulation although plants had more time to fight back?

It was worthy to check the effect on plant development and to check the expression of plant defensive genes in response to those nanoparticles. The authors could have used molecular and biochemical tools to prove their hypothesis.

Reply: We thank the referee for challenging us to bring forward experiments on this point. As a result, we have used a small panel of genes known to behave as indicators of biotic and abiotic stresses in arabidopsis. Responses to the three classes (but not all sizes) of nanoparticle were determined after 90 mins of treatment identical to that used for confocal microscopy. There was not a significant different response to neutral nanoparticles and so the hypothesis presented in the previous text was changed to reflect this result. However, there were significant responses to all the particle types, indicating that the plants do react rapidly. This more general message is now included in the text and the data are shown in Supporting Information. These new data do not change the conclusions or interpretations of the manuscript. We accept that full transcriptome responses are of interest and RNAseq might reveal charge-specific responses. However, this would be a considerable, costly and time-consuming operation that is beyond the scope of the current work. We believe that it should be amongst the topics of future work.

Authorship:

Given the request by Ref #2 to add new data, we needed to use additional techniques not familiar in our research groups. Therefore, Prof Napier secured the assistance of PhD students Chitra Joshi and Beth Richmond with the supervision and guidance of Professor Miriam Gifford (University of Warwick, UK) for the quantitative PCR data. In recognition of their contributions these names have been included in the revised author list.

Richard Napier
September 2022

REVIEWERS' COMMENTS

Reviewer #2 (Remarks to the Author):

The authors have addressed all my comments and concerns. I do not have further comments to make on the paper. This study is very interesting and worth publishing.